# Comparison of the Acute Effects of Carbohydrate Mouth Rinse and Coach Encouragement on Kinematic Profiles During Small-Sided Games in Young Male Soccer Players

**DOI:** 10.3390/nu17030546

**Published:** 2025-01-31

**Authors:** Yakup Zühtü Birinci, Serkan Pancar, Yusuf Soylu

**Affiliations:** 1Faculty of Sports Sciences, Bursa Uludağ University, Bursa 16059, Türkiye; 2Faculty of Sports Sciences, Aksaray University, Aksaray 68100, Türkiye; serkanpancar@aksaray.edu.tr; 3Faculty of Sports Sciences, Tokat Gaziosmanpasa University, Tokat 60250, Türkiye; yusuf.soylu@gop.edu.tr

**Keywords:** coaching behavior, soccer game-based training, nutritional strategies, verbal feedback

## Abstract

**Background**: Carbohydrate mouth rinsing (CHOmr), a nutritional intervention for delaying fatigue and meeting the energy demands of soccer, and the motivational strategy of coach encouragement (CE) are widely recognized as effective approaches for enhancing athletic performance in soccer. **Objectives:** This study aimed to compare the effects of CHOmr + CE, CHOmr, and CE on heart rate (HR) and kinematic profiles during four-a-side small-sided soccer games (SSGs). **Methods:** Twenty-four young soccer players (age: 17.2 ± 0.8 years) played six bouts of four-a-side SSGs with CHOmr + CE, CHOmr, or CE at 3-day intervals in a randomized, single-blinded, placebo-controlled, or crossover study design. The HR and kinematic responses were continuously recorded during all games. **Results:** There were no statistically significant differences between the groups in peak heart rate (HR_peak_) (*p* ≥ 0.05, F = 0.326, *p* = 0.723, η^2^ = 0.014) and mean heart rate (HR_mean_) (*p* ≥ 0.05, F = 0.845, *p* = 0.436, η^2^ = 0.035). No significant differences were found for distances in Zone 1 (*p* ≥ 0.05, F = 1.21, *p* = 0.306, η^2^ = 0.050), Zone 4 (*p* ≥ 0.05, F = 0.310, *p* = 0.735, η^2^ = 0.013), Zone 5 (*p* ≥ 0.05, F = 1.02, *p* = 0.368, η^2^ = 0.042), or Zone 6 (*p* ≥ 0.05, F = 0.161, *p* = 0.211, η^2^ = 0.055), nor acceleration (*p* ≥ 0.05, F = 0.208, *p* = 0.137, η^2^ = 0.083) and deceleration (*p* ≥ 0.05, F = 0.790, *p* = 0.460, η^2^ = 0.033). Similarly, although no significant differences were observed in the distance in Zone 3 (*p* ≥ 0.05, F = 3.12, *p* = 0.054, η^2^ = 0.119) or repeated sprint distance (*p* ≥ 0.05, F = 2.96, *p* = 0.062, η^2^ = 0.114), the CHOmr +CE group exhibited higher average values for these variables. However, a statistically significant difference was observed in the distance covered in Zone 2 (*p* ≤ 0.05, F = 3.89, *p* = 0.028, η^2^ = 0.145), with the CHOmr +CE group performing better, as confirmed by the post-hoc analyses. **Conclusions:** Although our findings indicate that CE alone may influence kinematic profiles during SSGs, similar to CHOmr or its combination with CE, further research should explore the underlying mechanisms and potential contextual factors influencing these outcomes. Therefore, we suggest that coaches prefer CE because it is easy to implement.

## 1. Introduction

Soccer is a team sport that requires athletes to meet technical and tactical demands. Because of its intermittent nature, soccer also requires athletes to simultaneously perform highly physically demanding actions without compromising the quality or quantity of their performance while meeting these requirements [1,2]. Soccer has become increasingly physically demanding over time, with an increase in the number and frequency of high-intensity actions, both with and without the ball, that play a crucial role in determining match outcomes [3]. Coaches must create a physically demanding environment that reflects game conditions during training to acquire adequately prepared players [4]. This has led sports scientists and coaches to develop training strategies that mimic games to optimize player performance and adapt to game demands.

Coaches widely prefer small-sided soccer games (SSGs) and game-based training methods that effectively replicate soccer-specific movements and patterns with fewer players and smaller pitch sizes [5,6]. The SSG training strategy aligns closely with the physical, technical, and tactical demands of soccer match play, enabling players to develop energetic and metabolic requirements simultaneously while engaging with the ball [7,8]. Numerous studies have shown that SSGs can be a valuable tool for enhancing training efficiency [9,10,11]. It is well established that manipulating structural and/or functional variables (e.g., format, pitch configuration, training regimen, rules, etc.) can cause different physiological and perceptual responses in players [12,13,14,15,16]. Therefore, coaches try to ensure that their players sustain maximum effort during training by adjusting the relevant training response to the “you train as you play and you play as you train” motto [17].

Verbal coach encouragement (CE) is an external motivational strategy that is widely implemented in a variety of sports training to enhance players’ performance. Research has demonstrated that CE can improve plenty of performance outputs such as upper-body power [18], maximal voluntary contraction [19], force production [20], balance [21], sprinting intensity [22], and overall physical fitness [23]. Furthermore, CE can be effective at improving performance in team sports such as soccer by providing positive athlete–environment interactions [24,25]. A study by Selmi et al. [26] compared SSGs with and without CE and demonstrated enhanced physical enjoyment and exercise intensity after four-a-side SSGs integrated with CE in young male soccer players. Similarly, Soylu et al. [27] showed that integrating CE with various forms of SSGs can increase psychophysiological responses and technical actions by beneficially affecting players’ focusing and engaging skills. Therefore, understanding the potential role of CE in boosting or sustaining players’ performance and preventing mental and physical fatigue during training can assist coaches in increasing the effectiveness of training sessions.

Given the evolving understanding of training methodologies, this study bridges this gap by exploring how carbohydrate (CHO) interventions, a key nutritional strategy for meeting the energetic demands of a soccer match [28], and coach encouragement, a motivational tactic, directly impact soccer players’ performance under match-like conditions. In recent years, as the evolution of modern soccer has resulted in more significant physical and mental fatigue [29], CHO interventions have become vital in delaying the onset of fatigue and maintaining performance [30]. Unlike the traditional consumption of CHO in solid, liquid, or gel forms, a practical alternative, CHO mouth rinsing (CHOmr), offers a more affordable and equally effective strategy by swishing a solution in the mouth for approximately 5–10 s and subsequently spitting it out [31,32]. Additionally, CHOmr provides a distinct advantage over CHO ingestion by reducing the risk of gastrointestinal distress in athletes during prolonged exercise [33], particularly in high-speed running where gastric emptying slows [34]. In this regard, many studies have investigated the ergogenic effects of CHOmr on soccer training. Although some studies [35,36] provided evidence supporting the positive impact of CHOmr on soccer-specific performance, others [37,38] have reported contradictory findings. Given the role of muscle and liver glycogen in supporting energy production and in tolerating fatigue during competition, highlighting the significance of using CHOmr as a nutritional strategy in meeting training goals could be important [39].

Thomson et al. [40] reported that mental and physical loads in soccer training, including fatigue, can be effectively managed by adjusting soccer-specific strategies. Despite extensive research on motivational and nutritional strategies to enhance soccer performance, a critical gap remains in understanding the comparative effectiveness of CE and CHOmr in SSGs. Although CE has shown promise in enhancing psychophysiological responses during SSGs, CHOmr offers a unique advantage in addressing physiological demands. However, to date, no study has directly compared these two approaches in the context of soccer training. Therefore, this study aimed to compare the effects of two strategies—CE as a motivational approach by coaches and CHOmr as a nutritional intervention—or their concurrent combination on kinematic profiles during SSGs. We hypothesized that both strategies yield distinct, synergistic benefits to kinematic profiles, with CE primarily influencing engagement and motivation and CHOmr enhancing physiological performance. By exploring these complementary approaches, this study seeks to provide novel insights into optimizing training methodologies, offering evidence-based guidance for coaches and sports scientists to improve player performance in soccer training.

## 2. Materials and Methods

### 2.1. Subjects

G*Power 3.1 software (G-Power, version 3.1.9.7, University of Düsseldorf, Düsseldorf, Germany) was used to determine the sample size using repeated-measures ANOVA (within-subject). Assuming a moderate effect size (f = 0.25), a significance level of 5% (α = 0.05), a statistical power of 80% (1-β = 0.80), a correlation of 0.6 between repeated measures, and sphericity (ε = 1), the minimum sample size required was 23 participants. A total of 24 amateur young male soccer players (age: 17.2 ± 0.8 years, height: 176.0 ± 5.0 cm, weight: 67.3 ± 4.5 kg, body mass index: 21.5 ± 0.7 kg/cm^2^; training experience: 7.2 ± 1.1 years) participated in six 4-a-side SSGs using either SSG_CE_, SSG_CHOmr_, or SSG_CHOmr + CE_ formats. SSGs were conducted at 72 h intervals to mitigate potential positive or negative effects on performance, as suggested in previous studies [13] (Figure 1). The participants fulfilled the following criteria: (i) a minimum of three years of soccer training, including four weekly sessions (≈90 min) and one match; (ii) absence of injuries during or in the month preceding the study; and (iii) abstention from drugs or performance enhancers throughout the study. Before giving written consent, the players and their parents received information about safety measures, potential risks, and assessment procedures. The primary objective of this study was not fully disclosed to prevent influencing outcomes. All the participants confirmed that they had no prior experience with the practice and were unaware of its benefits. This study was approved by the Research Ethics Committee and adhered to the guidelines of the Declaration of Helsinki.

### 2.2. Study Design

This study used a randomized, single-blind, and crossover design to compare the acute effects of three interventions (combined CHOmr and CE [CHOmr + CE], CHOmr, and CE) on heart rate (HR) responses and kinematic profiles of young male soccer players during SSGs. Before the interventions, participants completed the Yo-Yo Intermittent Recovery Test Level 1 (YYIRT-1) and were split into two groups based on aerobic fitness to ensure balanced teams for SSGs. Following a 15 min standardized warm-up involving jogging, sprinting, and soccer-specific movements, players engaged in a 4-a-side SSG format on an artificial grass field (Figure 1).

After the warm-up, the participants were instructed to swish 25 mL of 6.4% maltodextrin solution (CHOmr) for 10 s before spitting it back into the cup. The sequences of the SSGs (CHOmr + CE, CHOmr, and CE) were randomly determined (available online: www.randomizer.org). Each game consisted of six rounds of 4-a-side SSGs, with a 1-week interval between games. All SSG sessions were conducted simultaneously (09:30–11:30) to eliminate the effects of circadian variation on the variables. The air temperature was maintained between 28 and 30 °C, and the humidity was between 33% and 35%. Physiological responses and movement patterns were continuously monitored and recorded throughout the SSGs. The players received no tactical instructions or specific rules during the games, and multiple balls were positioned around the field to maintain continuous play. The coaches provided verbal encouragement for maximum effort throughout CHOmr + CE and CE, except for the CHOmr intervention. The standardized CE protocol encompasses reactive and spontaneous verbal behaviors, including technical instructions, corrections, and positive reinforcement (e.g., “good job!” or “well done”) as used in a previous study by Soylu et al. [27].

### 2.3. Procedures

#### 2.3.1. Anthropometric Measurements

Before breakfast, the players’ weight and height were measured using a body composition analyzer (BC-418MA, Tanita Corp., Tokyo, Japan). Using the bioelectrical impedance method, this analyzer uses multiple frequencies (ranging from 1 kHz to 50 kHz) to perform detailed body composition measurements.

#### 2.3.2. Yo-Yo Intermittent Recovery Level 1 Test

The participants performed the Yo-Yo Intermittent Recovery Test Level 1 (YYIRTL-1), a progressively challenging and acoustically guided evaluation to assess aerobic capacity [41], following the protocol outlined by Bangsbo et al. [42]. The HR of the participants was monitored throughout the test using a Polar V800 device (Polar OY; Kempele, Finland). The peak HR observed during YYIRTL-1 was recorded as the maximum HR. Upon completion of the test, the researchers estimated the maximal oxygen consumption (VO2max) using a predetermined formula: VO2max = 36.4 + (0.0084 × covered distance in YYIRT-1).

#### 2.3.3. Kinematic Profiles

Heart rate responses were tracked using a Polar H10 monitor. Simultaneously, player movements were precisely captured with a portable GNSS device equipped with a high-speed accelerometer (STATSports, Apex, Londonderry, Northern Ireland, version 2.0.2.4) to ensure accurate and comprehensive data collection. GNSS Apex devices were attached to the upper back between the shoulder blades, measuring 30 mm wide and 80 mm tall, and weighing 48 g each. These units showed exceptional reliability in tracking short distances and intermittent running [43]. Data from the GNSS units were extracted and examined using STATSports Apex software (Apex 10 Hz). For movement analysis, researchers selected six predefined speed categories—standing (0–0.7 km/h^−1^), walking (0.7–7.2 km/h^−1^), jogging (7.2–14.4 km/h^−1^), running (14.4–19.8 km/h^−1^), high-speed running (19.8–25.1 km/h^−1^), and sprinting (>25.1 km/h^−1^)—along with additional variables related to movement characteristics, based on previous studies [44].

#### 2.3.4. Supplementation Procedures

This study employed a single-blinded design. One group received a 6.4% maltodextrin solution prepared using Protein Ocean (Ankara, Türkiye). To ensure blinding, the supervising investigator added a non-caloric artificial sweetener, sucralose, in equal amounts to both solutions. Both solutions were prepared daily and maintained at room temperature. Each participant rinsed their oral cavity with 25 mL of their assigned solution in a pre-weighed plastic container for ten seconds. The container and its contents were weighed before and after each rinse to assess the potential solution retention. The participants were not informed about the effects of the maltodextrin intervention or the primary objective of the study until the conclusion of the study.

### 2.4. Statistical Analyses

The results of the current study are expressed as the mean ± standard deviation. The Shapiro–Wilk test was used to evaluate the data distribution. To examine the interactions and main effects on physiological responses and kinematic profiles, one-way repeated-measures analysis of variance (ANOVA) was conducted. For each independent variable, effect sizes were determined using Cohen’s d. Cohen’s d values were interpreted as follows: trivial (<0.20), small (0.20–0.59), moderate (0.6–1.19), large (1.2–1.99), and very large (≥2.0) [45]. The difference between the mean values of the measured variables was represented by the 95% Confidence Interval (95% CI). Statistical analyses were performed using SPSS software (version 27.0; SPSS Inc., Chicago, IL, USA). The threshold for statistical significance was set at *p* ≤ 0.05. Figures were visualized using the JASP 0.19.3 program.

## 3. Results

As shown in Figure 2, there was no statistically significant difference between the groups in HRpeak (F = 0.326, *p* = 0.723, η² = 0. 014) or HRmean (F = 0.845, *p* = 0.436, η² = 0.035).

Figure 3 demonstrates no statistically significant difference between the groups for Zone 1 (F = 1.21, *p* = 0.306, η^2^ = 0.050), indicating that CHOmr and/or CE did not influence the distances performed at low speeds. The results were similar for Distance Zone 4 (F = 0.310, *p* = 0.735, η^2^ = 0.013), Distance Zone 5 (F = 1.02, *p* = 0.368, η^2^ = 0.042), Distance Zone 6 (F = 0.161, *p* = 0.211, η^2^ = 0.055), acceleration (F = 0.208, *p* = 0.137, η^2^ = 0.083), and deceleration (F = 0.790, *p* = 0.460, η^2^ = 0.033). However, statistically significant differences were observed between the groups for Distance Zone 2 (F = 3.89, *p* = 0.028, η^2^ = 0.145); compared to the other groups, the CHOmr group performed better in these parameters according to post hoc analyses. Although there were no significant differences in Distance Zone 3 (F = 3.12, *p* = 0.054, η^2^ = 0.119) or repeated sprint distance (F = 2.96, *p* = 0.062, η^2^ = 0.114), the CHOmr group showed higher averages than the other groups for these variables.

## 4. Discussion

This study aimed to compare the effects of CE and CHOmr or CHOmr + CE interventions on kinematic profiles of four-a-side SSGs in young male soccer players. To our knowledge, this is the only study to compare CHOmr interventions and CE during game-based soccer training. Our results indicated that there was no significant difference between the training modification CE and nutritional strategy CHOmr intervention or CHOmr + CE on HR responses and distance in speed zones (1: standing, 2: walking; 3: jogging, 4: running, 5: high-speed running, and 6: sprinting), repeated sprints, acceleration, and deceleration during SSGs. There was only a significant difference in distance in Speed Zone 2 (walking), in which the CHOmr + CE group showed a greater distance.

Since the pioneering study by Carter et al. [46] demonstrated the potential beneficial effect of CHOmr intervention on 1 h cycling endurance exercise performance, many studies have replicated these effects in endurance sports [47,48,49]. Despite its potential benefits, the influence of CHOmr on SSGs remains largely unexplored in the current literature. To date, contradictory evidence has been derived from a few studies. A randomized controlled study by Soylu et al. [27] indicated that CHOmr intervention enhanced acute performance in intermittent and continuous forms of four-a-side SSGs in adolescent male soccer players. In contrast, Přibyslavská et al. [50] showed no improvement in anaerobic performance (including single or repeated jump tasks, short sprints, or shuttle running) among female collegiate athletes in an overnight fasting condition after a three-a-side SSG. A plausible reason for these findings is that multiple variables, such as the exercise protocol, fasting state, CHO solution characteristics (e.g., concentration and temperature), rinse duration, and placebo-related effects, may influence the ergogenic impact of CHOmr [51].

It is widely recognized that coaches can manipulate the mental and physical demands of soccer training to ensure that they meet specific objectives by adjusting strategies such as technical and tactical limitations [25,34]. In addition, Dixon et al. [52] indicated that coaches’ behavior has the potential to affect training outcomes, making it a key component of soccer-specific training strategies. In this regard, coaches mainly prefer CE to foster a supportive and challenging environment that promotes higher effort levels, sustains focus, and encourages optimal physical and mental responses from players [53,54,55]. Rampinini et al. [56] revealed that CE effectively increases exercise intensity (HR, blood lactate concentration [La], rated perceived exertion [RPE]) during SSGs among amateur soccer players. Similarly, Sampaio et al. [57] detected a significant increase in the RPE during two-a-side and three-a-side SSGs with CE, although there was no significant change in the percentage of HR_max_. In line with these findings, Selmi et al. [26] further demonstrated that CE enhances both the intensity of SSGs and the physical enjoyment of U-16 male soccer players. Some studies have evaluated the effects of verbal CE during SSGs implemented in physical education. Sahli et al. [58] demonstrated greater physiological responses, RPE, enjoyment, and positive mood by integrating CE during four-a-side SSG among 16 male school students. Similarly, Hammami et al. [24] reported that CE effectively increased female adolescent students’ performance on physical fitness tests and their physiological and technical responses during school-level SSGs. These studies highlight the key roles of CE in elevating the intensity of training sessions and modulating players’ psychophysiological responses during SSGs [59].

An interesting finding of our study was that we did not detect a potential synergistic effect from the sequential application of CE and CHOmr, probably because both methods are thought to have similar effects on motor performance. Anzak et al. [60] found that auditory stimuli (CE in this study) may allow motor pathways to be optimized beyond what can be achieved by voluntary effort alone. Similarly, CHOmr has been reported to mediate the central nervous system (elevated cortical activity in regions specifically involved in motor planning and motor command) by signaling muscles to perform efficiently over prolonged periods, thereby reducing perceived fatigue and sustaining muscular function [51]. In addition, studies have reported that feedback, such as CE [61] and CHOmr [62], may be associated with the activation of the striatum, which is widely regarded as a key brain region for processing rewards and is a system linked with motivation and perceived exertion. Many studies have explored the neural pathways underlying the mechanisms of CHOmr [63,64,65]. CHOmr activates fuel-sensing receptors in the mouth, likely mediated by taste receptors located in the papillae of the tongue and extending across the soft palate and larynx, triggering neural pathways that signal energy availability to brain areas such as the prefrontal cortex, orbitofrontal cortex, insula, and operculum frontal before digestion occurs. This afferent signaling enhances corticomotor excitability, facilitating motor output to both fresh and fatigued muscles. The mechanism likely involves sensorimotor integration via brainstem nuclei, leading to immediate ergogenic effects, even without CHO ingestion. These effects are more pronounced under fatigue conditions, highlighting the role of oral CHO receptors in modulating motor performance and emotional responses (perceived exertion). In addition, Chambers et al. [62] proposed that CHOmr (rinsing mouth with solutions containing glucose or maltodextrin) may reduce central fatigue by influencing dopamine pathways in the basal ganglia, thereby stimulating the reward circuitry and motor function areas, which could enhance motor function and increase the perception of effort. Similarly, CE may enhance motor performance by facilitating attentional focus and engaging the reward and motivational system. The ventral striatum, a key component of the basal ganglia, plays a crucial role in motivating force production, suggesting that attentional mobilization in response to encouragement is mediated by these neural circuits [66]. Given these findings, overlapping neural mechanisms may produce analogous responses, potentially explaining the lack of a synergistic effect observed in this study.

Another possible explanation that could affect the results of this study is the lack of a consistent and standardized protocol for CE. Hicheur et al. [67] highlighted that augmented feedback in soccer training might influence players’ perception of their coach as an evaluator, potentially increasing mental stress and cognitive load. This increased mental burden may disrupt the training environment and diminish its effectiveness. Conversely, the high intrinsic motivation and skill level of athletes in the present study may have limited the ability of CE to optimize arousal [68]. Based on the inverted-U curve theory [69], these situations may have reduced the optimum effect of the CE in this study.

Furthermore, the performance improvements gained from CE, through neural stimulation or delay of fatigue, may also depend on the individual characteristics [70]. Puce et al. [71] demonstrated that CE had a greater effect in swimmers with less experienced swimmers during a middle-distance event (200 m). Conversely, in our study, we recruited highly experienced young players; CE might have a decreased impact on this population.

Finally, in this study, we could not control or limit the athletes’ daily dietary routines determined by the team’s nutritionist due to it being a competitive period. This may have led to a variation in the extent of CHOmr’s influence on exercise performance. A recent review study revealed that CHOmr tends to elicit greater ergogenic effects in the fasting than the fed state [51].

Despite the cumulative knowledge derived from SSG studies, our understanding of the effects of CHOmr and CE on triggering kinematic demands during SSGs is still limited. We assume that the consequences of CE and CHOmr interventions on kinematic profiles would contribute to the design of SSGs with more specific game-based drills to meet competitive demands.

### 4.1. Practical Applications

This study revealed that the nutritional strategy of CHOmr and the motivational approach of CE showed no difference in kinematic responses during the four-a-side SSG. The major strengths of the current study are that it compares CHOmr and CE, which are widely used, especially in soccer training, in a practical manner. Given the similar effects of these strategies on the kinematic responses of SSGs, it is recommended that CE, which is highly advantageous in terms of practicality, should be preferred over CHOmr for integration into SSGs to meet the specific training goals or player needs of coaches and practitioners. In addition, coaches should consider the social aspects of their behavior, as players’ perceptions of coaching can play a crucial role in enhancing their sense of competence and intrinsic motivation during games. These findings offer practical insights for coaches to optimize players’ well-being during training sessions.

Although this study did not observe significant distinct effects from between CHOmr and CE, the well-established benefits of CHOmr in delaying fatigue suggest that it may be particularly useful during high-intensity or prolonged training sessions, where energy demands are high. CHOmr may help maintain higher performance levels, particularly in the later stages of training sessions. In situations where player–coach or player–player dynamics are disrupted, or during compensatory training for players with low arousal levels due to limited match time, the effectiveness of verbal coach encouragement may be diminished. In such cases, implementing carbohydrate interventions may be especially beneficial.

### 4.2. Limitations

To address the limitations of the current study, it is essential to acknowledge that the limitations and various variables may have influenced the results. In this study, we focused only on young, well-trained, and motivated male soccer players, which limits the generalizability of our findings to competitive soccer players from different divisions, female soccer players, or older players. Another key limitation of this study was the use of only a four-a-side SSG format. SSG configurations (number of players, individual area per player, width-to-length ratio, using or not using goals or goalkeepers, limitation in ball touches or movements, sets, repetitions, work-to-rest ratio, etc.) are well established. They can elicit different biological, physical, tactical, or technical responses.

Acknowledging the limitations of the participant group in this study, we suggest that future research should explore the effects of CHOmr and CE across a more diverse and broader population, which could enhance our understanding of the ergogenic effects of CHOmr and the motivational effects of CE.

## 5. Conclusions

In conclusion, this study provides valuable insights into the comparative effects of CHOmr and CE on kinematic responses during four-a-side SSGs. Despite both strategies having distinct beneficial effect in soccer training, our results indicated no significant differences in kinematic responses during SSGs between CE, CHOmr, and their combination, except for walking distance, which was greater in the CHOmr + CE group. These findings suggest that both strategies may exert overlapping influences on motor performance and psychophysiological responses, limiting potential synergistic effects when implementing combinations.

Given CE’s simplicity, cost-effectiveness, and ability to enhance players’ focus and effort, it may serve as a preferred approach for enhancing training environments, especially when logistical or financial constraints limit the use of nutritional interventions like CHOmr. Future research should explore these interventions across diverse player groups such as female players, recreational or professional groups, game formats, and broader training contexts to refine and optimize their application in soccer practice.

## Figures and Tables

**Figure 1 nutrients-17-00546-f001:**
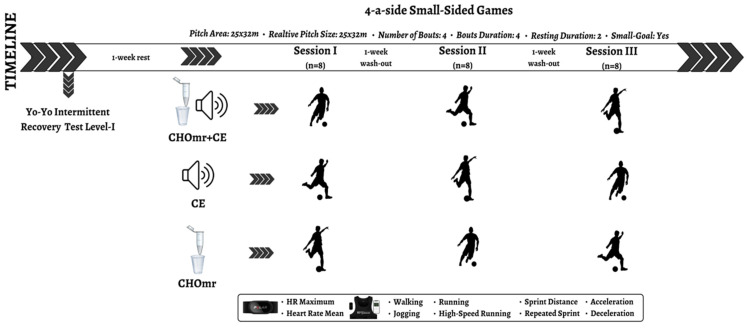
Study design of SSGs.

**Figure 2 nutrients-17-00546-f002:**
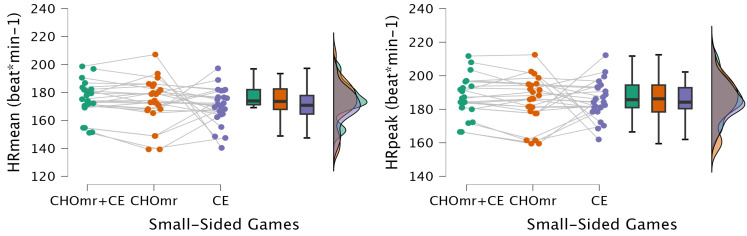
Heart rate (HR) responses (CHOmr+CE: combined carbohydrate mouth rinse and coach encouragement; CHOmr: carbohydrate mouth rinse; CE: coach encouragement).

**Figure 3 nutrients-17-00546-f003:**
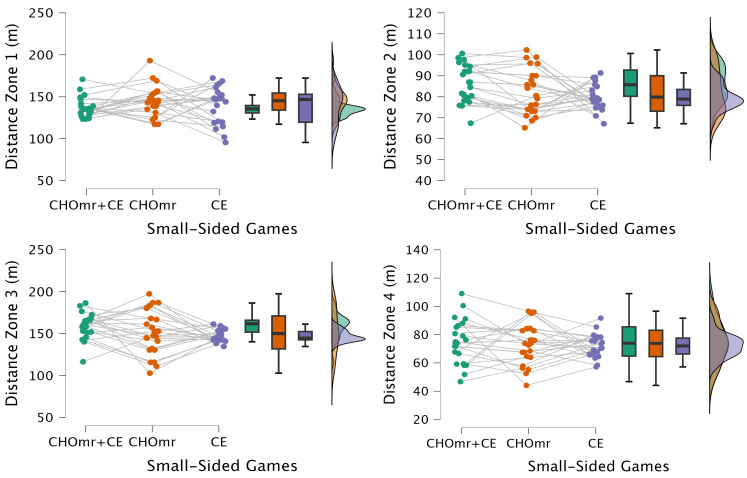
Kinematic profile (CHOmr: combined carbohydrate mouth rinse and coach encouragement; CHOmr: carbohydrate mouth rinse; CE: coach encouragement; ACC: accelerations: DCC: decelerations).

## Data Availability

The data are available from the corresponding author upon reasonable request.

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
