# Peer review of "Comparison of the Acute Effects of Carbohydrate Mouth Rinse and Coach Encouragement on Kinematic Profiles During Small-Sided Games in Young Male Soccer Players"

_nutrients, 2025, doi:10.3390/nu17030546_

Round 1

Reviewer 1 Report

Comments and Suggestions for Authors

Dear Authors,

The article submitted for consideration to “Nutrients” Journal is well written and scientifically grounded. The methodology is robust, the randomized crossover design minimizes inter-individual variability, ensuring a fair comparison of interventions. Finally, the use of standardized protocols, such as the Yo-Yo Intermittent Recovery Test, enhances the validity and reproducibility of the results. My suggestion is to add in the conclusion section the Authors’ hypothesis in applying these methods to different population groups, such as female players or older athletes. Moreover, the Authors should better highlight and explain that a lack of standardized verbal encouragement protocols might influence outcomes.

Reviewer 2 Report

Comments and Suggestions for Authors

paper - Comparison of the Acute Effects of Carbohydrate Mouth Rinse and Coach Encouragement on Kinematic Profiles During Small-Sided Games in Male Young Soccer Players

here are some recommedations, questions, suggestions, and comments

paper is well written - abstract concise and clear

in the abstract - would suggests to clarify more the ... carbohydrate mouth rinsing (CHOmr) or  carbohydrate (CHO) interventions; a nutritional strategy for meeting the energetic demands of a soccer match and the motivational impact of coach encouragement (CE) are widely used and beneficial ...

this would further clarify what CHOmr is

methodology is also well written, togethe with the results and discussions

could help to clarify - why 72 hours interval ....

authors mentioned - 72-hour intervals to mitigate potential positive or negative ef-fects on performance - citation is needed

just curious would the 4-a-side small-sided games be too simple, hence the no significant differences?

or typical in experimental study design the participants might know they are being tested (hence perform better)?

(or since its double blinded as the author/s mentioned) the participants does not know they are being tested?

what now? provide more practical implications, besides encouraging couaches to use verbal encouragement
